# Intrinsic donor-bound excitons in ultraclean monolayer semiconductors

Pasqual Rivera [1], Minhao He [1], Bumho Kim[2], Song Liu[2], Carmen Rubio-Verdú[3], Hyowon Moon[4], Lukas Mennel[4], Daniel A. Rhodes[2], Hongyi Yu[5], Takashi Taniguchi [6], Kenji Watanabe [7], Jiaqiang Yan[8,9], David G. Mandrus[8,9,10], Hanan Dery[11], Abhay Pasupathy [3], Dirk Englund [4], James Hone [2✉], Wang Yao [5✉] & Xiaodong Xu [1,12✉]

The monolayer transition metal dichalcogenides are an emergent semiconductor platform exhibiting rich excitonic physics with coupled spin-valley degree of freedom and optical addressability. Here, we report a new series of low energy excitonic emission lines in the photoluminescence spectrum of ultraclean monolayer WSe$_2$. These excitonic satellites are composed of three major peaks with energy separations matching known phonons, and appear only with electron doping. They possess homogenous spatial and spectral distribution, strong power saturation, and anomalously long population lifetimes (>6 μs) and polarization lifetimes (>100 ns). Resonant excitation of the free inter- and intravalley bright trions leads to opposite optical orientation of the satellites, while excitation of the free dark trion resonance suppresses the satellites' photoluminescence. Defect-controlled crystal synthesis and scanning tunneling microscopy measurements provide corroboration that these features are dark excitons bound to dilute donors, along with associated phonon replicas. Our work opens opportunities to engineer homogenous single emitters and explore collective quantum optical phenomena using intrinsic donor-bound excitons in ultraclean 2D semiconductors.

[1] Department of Physics, University of Washington, Seattle, WA 98195, USA. [2] Department of Mechanical Engineering, Columbia University, New York, NY 10027, USA. [3] Department of Physics, Columbia University, New York, NY 10027, USA. [4] Department of Electrical Engineering and Computer Science, Massachusetts Institute of Technology, Cambridge, MA 02139, USA. [5] Department of Physics, University of Hong Kong, and HKU-UCAS Joint Institute of Theoretical and Computational Physics at Hong Kong, Hong Kong, China. [6] International Center for Materials Nanoarchitectonics, National Institute for Materials Science, Tsukuba, Ibaraki 305-0044, Japan. [7] Research Center for Functional Materials, National Institute for Materials Science, Tsukuba, Ibaraki 305-0044, Japan. [8] Materials Science and Technology Division, Oak Ridge National Laboratory, Oak Ridge, TN 37831, USA. [9] Department of Materials Science and Engineering, University of Tennessee, Knoxville, TN 37996, USA. [10] Department of Physics and Astronomy, University of Tennessee, Knoxville, TN 37996, USA. [11] Department of Electrical and Computer Engineering, University of Rochester, Rochester, NY 14627, USA. [12] Department of Materials Science and Engineering, University of Washington, Seattle, WA 98195, USA. ✉email: jh2228@columbia.edu; wangyao@hku.hk; xuxd@uw.edu

A promising route for optical encoding of matter is to employ excitons, Coulomb-bound electron-hole pairs, which are elementary optical excitations in semiconductors. Monolayer transition metal dichalcogenides (TMDs) are an emergent platform for exploring excitonic physics at the two-dimensional (2D) limit. This is largely due to their strong light-matter interactions, easy access to electric and magnetic control, and unique combination of spin-valley coupling and valley contrasting circular dichroism[1–4]. Encapsulation of monolayer TMDs within hexagonal boron nitride (hBN) has led to drastically improved sample quality[5,6], allowing the identification and detailed studies of a variety of optically bright and dark valley excitonic states[7–21], and their phonon replicas[22,23] assisted by both zone center[24,25] and zone edge phonons[26,27].

Despite the rapid progress in sample quality and understanding of their excitonic physics, these monolayer semiconductors are still far from perfect, and questions remain. For example, while localized single-photon emitters have been observed in these materials[28–31], they exhibit random emission energies over a broad spectral range (>100 meV), and commonly lose the desirable valley optical selection rules due to the random anisotropy. Moreover, they generally appear in low quality samples, where inhomogeneous broadening obscures the underlying rich excitonic manifold that has been observed in clean samples. While the precise nature of these quantum light sources remains unclear, O interstitials[32,33] and extrinsic confinement potentials such as strain appear to play a role in the localized emission[34–38]. However, even the cleanest samples reported to date inevitably contain intrinsic defects (e.g. self-flux growth WSe$_2$ crystal with defect density of ~$1 \times 10^{11}$ cm$^{-2}$)[39]. Several of these native defects have been identified and their electronic structure probed using scanning tunneling microscopy (STM) and spectroscopy[40–43]. A natural question arises as to if new spectral features may arise from excitons bound to such dilute intrinsic defects in these ultraclean monolayer crystals[44].

In this work, we report the observation of donor bound dark excitons and their phonon replicas in ultraclean monolayers of WSe$_2$ encapsulated in hBN. The samples in our study ($N = 10$) were fabricated from different sources of WSe$_2$ and hBN bulk crystals and yielded consistent and reproducible results. The data presented in the main text are mainly from two devices. For electrostatic control of the charge carrier density in the WSe$_2$, a local graphite bottom gate is used. Fig. 1a, b show an optical

microscope image of Device 1 and its schematic, respectively. The following experiments were performed at a temperature of either 1.6 or 4 K, and with excitation laser of energy 1.96 eV, unless otherwise specified.

## Results

**Observation of robust excitonic satellites**. The PL intensity of Device 1 as a function of back gate voltage ($V_b$) and emission energy is shown in Fig. 1c. The excitation power is 5 μW. All reported excitonic features such as excitons, trions, dark states, and various phonon replicas in the spectral range of 1.65 to 1.75 eV are well-resolved (Supplementary Fig. 1). The sample is also nearly intrinsic, with charge neutrality occurring at $V_b \approx -0.25$ V. All these factors attest to the high sample quality. An observation is the emergence of low energy excitonic satellites occurring between 1.58 and 1.63 eV under n-type doping. The three main features, indicated as S1, S2, and S3 from high to low energy, are the focus of this paper.

The excitonic satellites have homogeneous spatial distribution and energy across the entire sample. Figure 1d displays the spatial map of the satellite emission intensity (spectral range shown on top of Fig. 1c at $V_b = 0.5$ V), while Fig. 1e shows the spatial distribution of S1 peak energy. Its peak position relative to neutral exciton is unchanged across the whole sample (See Supplementary Fig. 2). Remarkably, the spectral structure and energetic positions of the excitonic satellites are also consistent across many samples. The normalized PL spectrum of three exemplary samples under similar experimental conditions is shown in Fig. 1f. The satellite emission energies, relative to the neutral exciton, between total 10 different samples fabricated over a 3-year span with different crystal sources are well aligned (Supplementary Fig. 3). From the spatial and energetic homogeneity of the satellite PL, we can safely rule out their origin from contamination, cracks, or other extrinsic confinement potentials.

**Evidence of intrinsic defect-bound excitons**. The satellite emission exhibits distinct power dependence compared to the known free 2D excitons. Figure 2a presents the normalized PL spectra of Device 2 with powers ranging over three orders of magnitude (20 nW to 60 μW). The satellite emission dominates the entire spectrum at low powers, but quickly saturates and the free 2D excitonic manifold appears as the power increases. Above

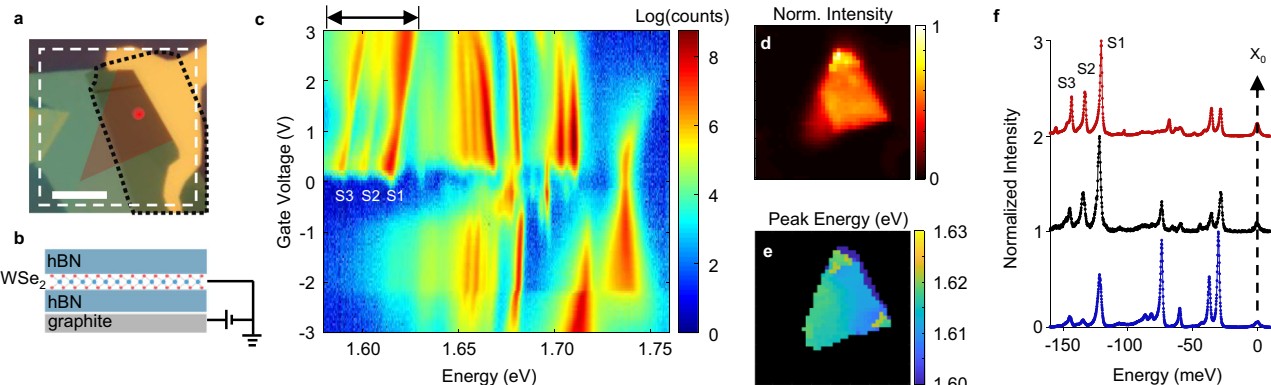

**Fig. 1 Emergence of deeply bound exciton satellites. a** Optical image of electrostatic gating device composed of exfoliated monolayer WSe$_2$ (red area) encapsulated in hBN with graphite backgate (outlined in black). Scale bar is 10 μm. **b** Schematic of sample side view. **c** Photoluminescence as a function of back gate voltage with the laser spot indicated by the red dot in **a**. Three satellites peaks near 1.60 eV appear when the sample is n-doped. **d** Spatial map of the integrated PL from the satellite peaks at $V_b = 0.5$ V. Integration spectral region shown by arrows on top of **c**, and spatial region outlined by dashed white lines in **a**. **e** Spatial map of the peak energy of the highest-energy satellite at $V_b = 0.5$ V. **f** Waterfall plot of PL spectra from three different samples, showing homogeneous satellite binding energies and robust three peak spectral features. The energy axis is scaled relative to the free neutral exciton ($X_0$ at ~1.735 eV).

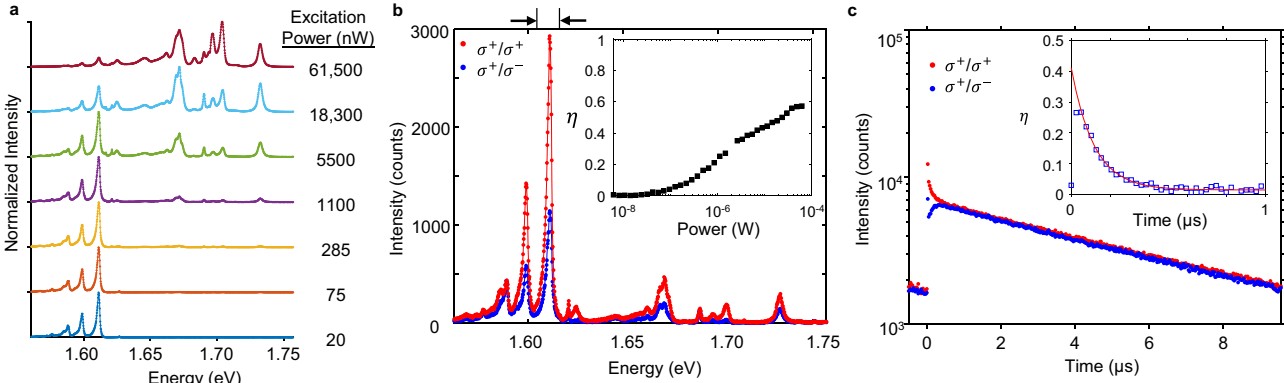

**Fig. 2 Power, polarization, and time-resolved PL from satellites. a** Waterfall plot of normalized PL spectra at selected excitation powers. The exciton satellites dominate the spectrum at low powers. **b** Polarization-resolved spectrum of satellite peaks under $\sigma^+$ circularly polarized excitation (1 μW). Inset: power dependence of $\eta$, the degree of circular polarization, which grows with increasing power up to $\eta \approx 0.6$. **c** Time-resolved PL of S1 reveals a population lifetime of 6.95 ± 0.05 μs and a polarization lifetime of 116 ± 4 ns (red line is single exponential fit). All data are from Device 2 at $V_b = 0$ V. Top arrows in **b** indicate the spectral region of integration.

10 μW excitation power, the satellite emission is overwhelmed by the linear response of the higher energy 2D exciton species (see Supplementary Fig. 4 for additional samples). The strong power saturation is a hallmark of defect-localized excitons[45], and the low saturation threshold implies low defect density and long exciton lifetime. Note that the integrated photoluminescence saturation count rate is comparable to that of single-photon emitters[31].

Circularly polarized optical pumping reveals non-trivial polarization of the satellite PL. We observe that the two highest-energy satellites, S1 and S2, are co-circularly polarized with the excitation laser, while the lowest energy satellite S3 is nearly unpolarized, as shown in Fig. 2b for Device 2. The degree of circular polarization is defined as $\eta \equiv (I_+ - I_-)/(I_+ + I_-)$, where $I_\pm$ denotes the intensity of the $\sigma^\pm$ polarized components of the PL. For S1, we observe that $|\eta| \sim 0$ at low excitation power (<100 nW), but increases with the excitation power to a value of $|\eta| \approx 0.6$ at powers above ~10 μW (inset of Fig. 2b, integration over S1). Optical orientation of exciton spins is often observed in excitons bound to defects, where the localized electrons can become spin polarized, e.g. via efficient exchange interactions with photoexcited free electron spins[46]. The strong power dependence of circular polarization is thus another signature of localized excitonic spin states. We note that under linearly polarized excitation, the satellite emission is not polarized in the linear basis, regardless of the axis of excitation (Supplementary Fig. 5). The lack of linear polarization and preservation of the circular optical selection rules imply underlying rotational symmetry ($C_3$) of the defect. This behavior is different from all reported quantum emitters found in WSe$_2$, which emit with linear polarization with strong spatial and spectral inhomogeneity[28–31].

Time-resolved PL reveals anomalously long lifetime of the satellite states, as shown in Fig. 2c. A bi-exponential fit of the long PL component yields a population lifetime of 6.95 ± 0.05 μs. This is 3–5 orders of magnitude larger than that of both 2D bright and dark excitonic species[13,47,48], and consistent with the observed strong saturation of the satellite PL. We further extract the circular polarization lifetime of the satellite emission to be 116 ± 4 ns, as determined by fitting $\eta$ with a single exponential (inset of Fig. 2c). The polarization lifetime is also ~2–4 orders of magnitude longer than that of 2D bright[48] and dark excitons[41,42]. Such an enormously long population lifetime implies a dramatic reduction in the non-radiative lifetime of the satellite state. Moreover, the long polarization lifetime suggests small intervalley

scattering rate, and weak transverse-longitudinal splitting, which further supports high symmetry ($C_3$) of the underlying confinement center.

All the above experimental features confirm that the excitonic satellites have distinct origin from both localized quantum emitters and free excitonic species previously reported in monolayer WSe$_2$. The spatial and spectral homogeneity across many samples, unusually long population and polarization lifetimes, and strong pump power dependence in both emission intensity and polarization, imply their origin as excitonic states bound to dilute intrinsic defects. The strong dependence of the observed PL features on electron doping is the first indication of the defect type as donors.

**Controllable synthesis of crystals with intrinsic defect-bound excitons.** Figure 3a–c shows the wide field STM topographic images of WSe$_2$ with three different growth parameters (see Methods, labeled as F1, F2, and F3). Two main types of defects, bright and dark features in these images, are observed. Scan tunneling spectroscopy reveals in gap states, implying the bright and dark defects are donor and acceptor in nature, respectively (see Supplementary Fig. 6 and Supplementary Note 1 for details). Across samples, F1, F2, and F3, the total defect density increases from $2 \times 10^{11}$ to $4 \times 10^{11}$, and to $8 \times 10^{11}$ cm$^{-2}$, while the donor density decreases from $1.5 \times 10^{11}$ cm$^{-2}$, to $1.3 \times 10^{11}$ cm$^{-2}$, to $4 \times 10^{10}$ cm$^{-2}$. Figures 3d, e show the corresponding gate dependent photoluminescence of monolayers exfoliated from these three types of crystal. The excitonic satellites are clear in sample F1 with abundant donors but lowest total defect density, are barely visible in F2, and are not seen in sample F3. The satellites peak intensity thus track the donor density, supporting assignment of the observed photoluminescence satellites to excitonic states bound to dilute donor defects.

**Donor bound dark excitons and phonon replicas.** We examine the energy difference between the observed satellites peaks. The extracted energies of S2 and S3, relative to S1, at zero gate voltage are (12.5 ± 0.7) meV and (23.1 ± 0.5) meV, respectively, where the error bar is the standard deviation of the peak positions from 10 samples. These energy differences are similar to the calculated phonon energy of $K_2$ (12 meV) and $\Gamma_5$ (22 meV), respectively[26]. We thereby attribute S1 to the zero-phonon line of donor bound excitonic emission, while S2 and S3 as its $K_2$ and $\Gamma_5$ (or E′) phonon replicas. The assignment of S2 and S3 resembles

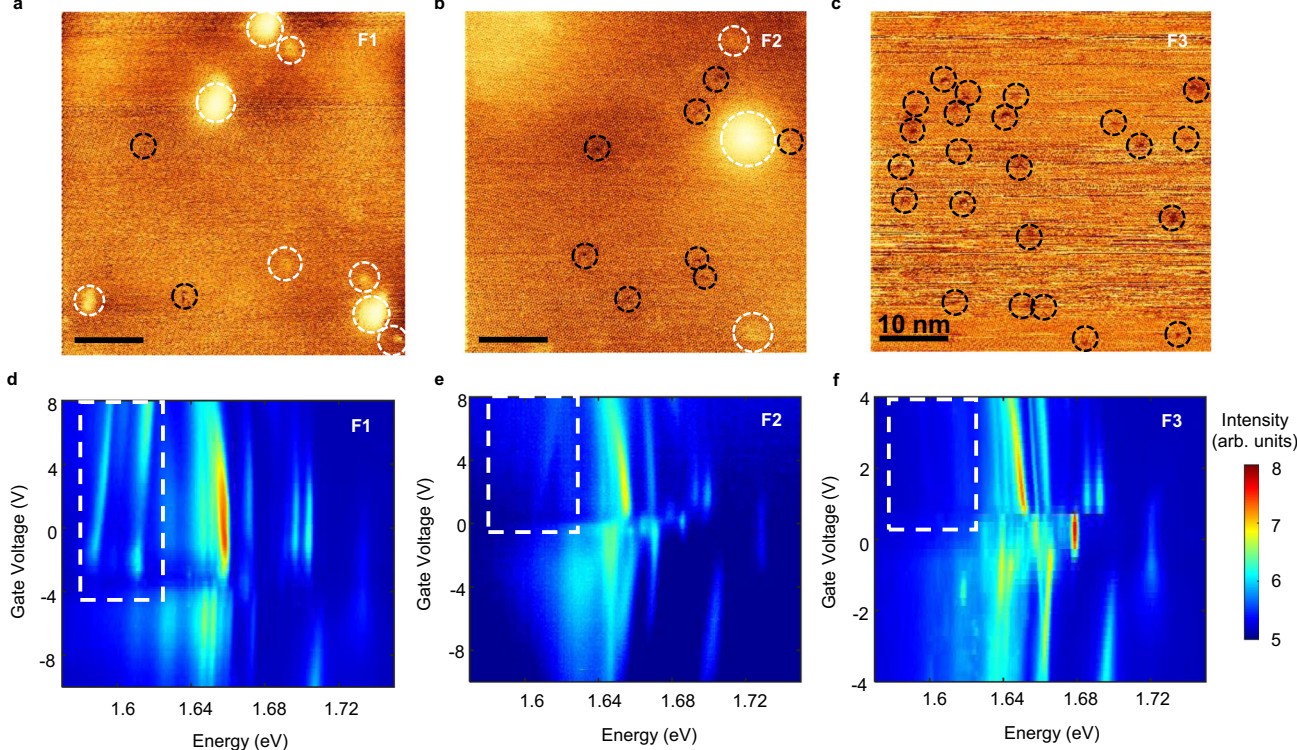

**Fig. 3 Characterization of defects and defect-bound excitons in WSe2. a–c** Scanning Tunneling Microscopy (STM) topographic images of 50 x 50 nm$^2$ area of WSe$_2$ with different growth parameters (F1, F2, and F3). Scale bar is 10nm. Imaging conditions for the STM topographic images were a tunneling bias of 1.4 V and current of 400 pA for F1 and a tunneling bias of 1.4 V and current of 150 pA for F2 and F3. The WSe$_2$ crystals were cleaved in ultra-high vacuum STM chamber (base pressure <2.0 × 10$^{-10}$ torr) to obtain a clean surface before imaging. White and black dashed circles highlight the donor and acceptor defects, respectively. See Supplementary Fig. 6 details. **d–f** Photoluminescence (PL) color maps as a function of the back-gate bias for **d**, F1, **e**, F2, and **f**, F3. The white dashed boxes in the PL color maps indicate the energy and the back-gate bias ranges for defect-bound excitons. For the PL color map, the samples were excited using a continuous-wave (CW) laser with an excitation wavelength of 532 nm with a fluence of 650 W/cm$^2$ (10 μW) at 4 K.

the observed phonon replicas for free 2D excitons, pointing to the strong phonon-exciton interactions in monolayer WSe$_2$[24–27]. The E″ phonon replica from a localized quantum emitter have also been observed in monolayer WSe$_2$,[49] albeit with different emitter properties from those reported here.

Since in WSe$_2$ the ground state configuration of a free exciton is the spin-forbidden dark exciton $X_d$,[26,27] we infer that the zero-phonon line S1 arises from a donor bound dark exciton $DX_d$. The dark exciton long lifetime makes possible efficient formation of $DX_d$, even at very low exciton density. This state should be composed by a positive charged ion, a donor bound electron, and an electron–hole pair. For the lowest energy configuration, the two electrons should have opposite spins to form spin singlet state.

**Photoluminescence excitation spectroscopy.** The above understanding of the charge configuration of $DX_d$ is further supported by polarization-resolved PL excitation spectroscopy (PLE). We tuned the energy of a continuous wave laser from 1.65 to 1.75 eV while collecting the satellite emission. As a reference, the PL spectrum obtained with 1.96 eV excitation is provided in Fig. 4a. The polarization-resolved PL response of the satellites under $\sigma^+$ excitation is shown in the left ($\sigma^+$ detection) and middle ($\sigma^-$ detection) panels of Fig. 4b. The extracted PL polarization $\eta$ is shown in the right panel with the satellite PL under 1.96 eV excitation overlaid in black. Comparing Fig. 4a, b, we can see that satellite emission is greatly enhanced when the laser is in resonance with free neutral exciton, which produces large population of dark excitons and thus $DX_d$. The satellite emission is

suppressed when resonant with the dark trion. This can be understood as the creation of dark trion competes with the formation of dark exciton, and thus donor bound dark exciton states.

A notable feature is the polarization-dependent response when the laser is in resonance with free bright trions. As shown in the $\eta$ panel in Fig. 4b, the blue regions represent PL polarization reversal when the laser is resonant with the intervalley trion ($X_T^-$), opposite to that under excitation at the intravalley resonance ($X_S^-$). This contrast is highlighted by polarization-dependent spectra in Fig. 4c. The satellite emission is co- and cross-circularly polarized with laser in resonance with $X_S^-$ ($E = 1.699$ eV, bottom panel) and $X_T^-$ ($E = 1.706$ eV, top panel), respectively. Note that the vertical white stripe in $\eta$ at ~1.59 eV indicates that S3 has negligible circular polarization.

## Discussion

**Donor spin state initialization.** Figure 5 explains the above trion excitation dependent PL polarization via donor spin state initialization process (see Supplementary Fig. 7 and Supplementary Note 2 for details). Figure 5a depicts the valley-spin coupled band edges, with initially unpolarized electrons on shallow donor levels[50]. Using resonant excitation of intervalley trion $X_T^-$ in Fig. 5b as an example, $\sigma^+$ polarized excitation creates an electron and hole pair in the K valley, which pairs with a spin-up electron in the K′ valley and forms $X_T^-$. This optical pumping process depletes the spin-up electrons associated with donors in K′, eventually resulting in a net population of spin down electrons in the K valley bound to the shallow donors (denoted as $D_{K,\downarrow}$). $D_{K,\downarrow}$

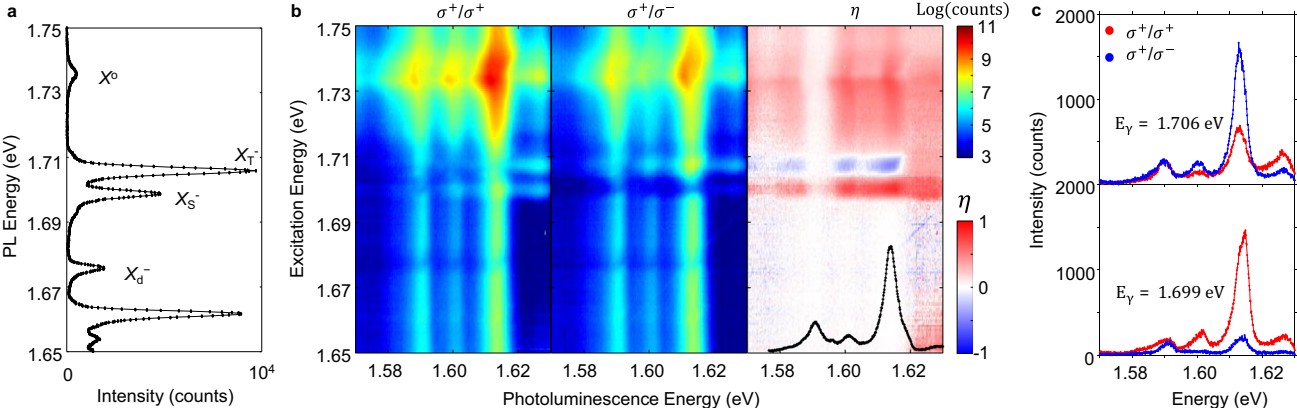

**Fig. 4 Excitation energy dependent spin-valley depletion and PL polarization reversal.** All data are from Device #1 at $V_b = 0.5$ V. **a** PL spectra with a HeNe laser excitation. **b** Polarization-resolved PL spectra by sweeping $\sigma^+$ polarized excitation from 1.75 to 1.65 eV. The $\sigma^+$ and $\sigma^-$ components of the PL are shown in the left and middle panels, respectively, while the right panel shows the extracted degree of polarization $\eta$. **c** Polarization-dependent PL with the excitation in resonance with the free intervalley (top, 1.706 eV) and intravalley (bottom, 1.699 eV) trions, showing polarization reversal of the exciton satellite PL.

then selectively binds to a dark exciton in the $K'$ valley ($X_d^{K'}$), because of the Pauli exclusion. The donor bound state emission can happen via either defect assisted direct electron-hole recombination yielding S1 PL peak, or phonon assisted stokes emission (either $K_2$ for S2 peak or $\Gamma_5$ for S3 peak), via an intermediate bright trion state (Fig. 5c). In all cases, the emission is $\sigma^-$ polarized as determined by the hole valley configuration, cross-polarized to the excitation[26].

When $\sigma^+$ polarized excitation is in resonance with intravalley trion $X_S^-$, it depletes instead the spin down electrons in $K$ valley, and leaves a net population of spin-up electron in $K'$ valley bound to the shallow donor ($D_{K',\uparrow}$), as shown in Fig. 5d. This leads to donor bound dark exciton states ($DX_d^K$) with spin-valley polarization opposite to that in resonant excitation of $X_T^-$. The donor bound state then emits $\sigma^+$ polarized light, co-polarized with the excitation (Fig. 5e). We note that, although pump is $I\sigma^+$ polarized, supply of free neutral dark exciton is expected in both valleys because of the ultrafast valley depolarization through electron–hole exchange in the formation of neutral excitons, so the $DX_d^K$ spin-valley polarization is determined by the optical orientation of donor electron. We would also like to point it out that although S2 behaves similarly to S1, which both have strong circular polarization, $S_3$ is unpolarized. This suggests S3 maybe not be as simple as a $\Gamma_5$ phonon replica of S1, which needs further investigation.

**Origin of intrinsic defect.** Our results demonstrated the observation of excitons bound to intrinsic defects. Extrinsic defects (e.g. O at Se site) and confinement potential, which can be responsible for localized single-photon emitters with strong optical anisotropy[32–38], are thus not candidates. There are four types of intrinsic defects: W vacancy, Se vacancy, W replace Se site, and Se replace W site. Although Se vacancy has been suggested to be responsible for observed broad low energy PL features[51], our atomically resolution STM measurements (Supplementary Fig. 6e) rules out vacancy as donor candidates. In addition, previous study on similar crystals with low defect density found the chalcogenide vacancy to be very rare[39]. Calculation also suggested Se at W antisite ($Se_W$) is a deep defect with multiple in gap states[32], and the exciton bound to $Se_W$ is expected to be about 300 meV below the $WSe_2$ A exciton.[32] All these are distinct from our experimental observation: our STS on donor (Supplementary Fig. 6a) only shows one shallow defect band near the CBM, while the donor bound exciton is about 120 meV below the A exciton. The leaves W at Se antisite ($W_{se}$) as a

possible candidate. However, calculation suggests that $W_{se}$ is an acceptor, which has multiple charge levels with a band near CBM[52]. This is distinct from our results that the defect is donor type with only one charge level within the gap. The current study cannot resolve the exact type of the defect, which requires further experimental and theoretical efforts (see discussion).

**Outlook.** Our results reveal light emission from dark exciton bound to intrinsic dilute donors and possible phonon replicas in the ultraclean monolayer $WSe_2$. Similar behavior is observed in III–V semiconductors[53], where defect density must be low before individual defect-bound exciton peaks and charging states are resolvable. It remains to be seen whether the defect-bound excitons exist in multilayer samples and other transition metal dichalcogenides. The observed long population and polarization lifetimes are advantageous for exploiting the spin-valley functionalities using monolayer $WSe_2$. The possibility of probing single-donor emitting sites by further reducing the native defect density, or by using near-field techniques, is an interesting direction towards realizing optical spin quantum memory in 2D materials. The PLE results are promising in this respect, since they demonstrate optical initialization of the donor-bound electron spin state by selective excitation of different trion states. On the other hand, interactions between neighboring donor sites may lead to interesting many-body behaviors, such as long-range correlation. Non-classical photon statistics are expected to emerge in both regimes, although weak oscillator strength may pose significant challenges. Our work prompts research efforts to identify and engineer the underlying defect and its electronic configuration, and we expect that further improvements in sample quality will enable detailed studies of the satellite fine structure, such as vibrational and rotational spectrum, hyperfine interactions, excited states, and optical orientation by magnetic fields control of donor spin states.

## Methods
**Crystal growth.** For growth of $WSe_2$ crystals with varying defect densities (Fig. 3), $WSe_2$ crystals were synthesized by reacting W with Se flux. W powder (99.999%) and Se shot (99.999%) mixed in 1:100 (F1), 1:15 (F2), and 1:5 (F3) atomic ratios, were loaded into quartz ampules separately which were then evacuated and sealed at ~$10^{-6}$ Torr. The ampules were vertically placed into a box furnace and heated to 1080 °C over 48 h. After a dwelling time of 1 week at 1080 °C, it was slowly cooled down to 300 °C at a rate of 0.6 °C/h. The obtained $WSe_2$ crystals were subsequently filtered from the Se flux by quartz wool and annealed at 275 °C for 24 h in a vacuum quartz ampule.

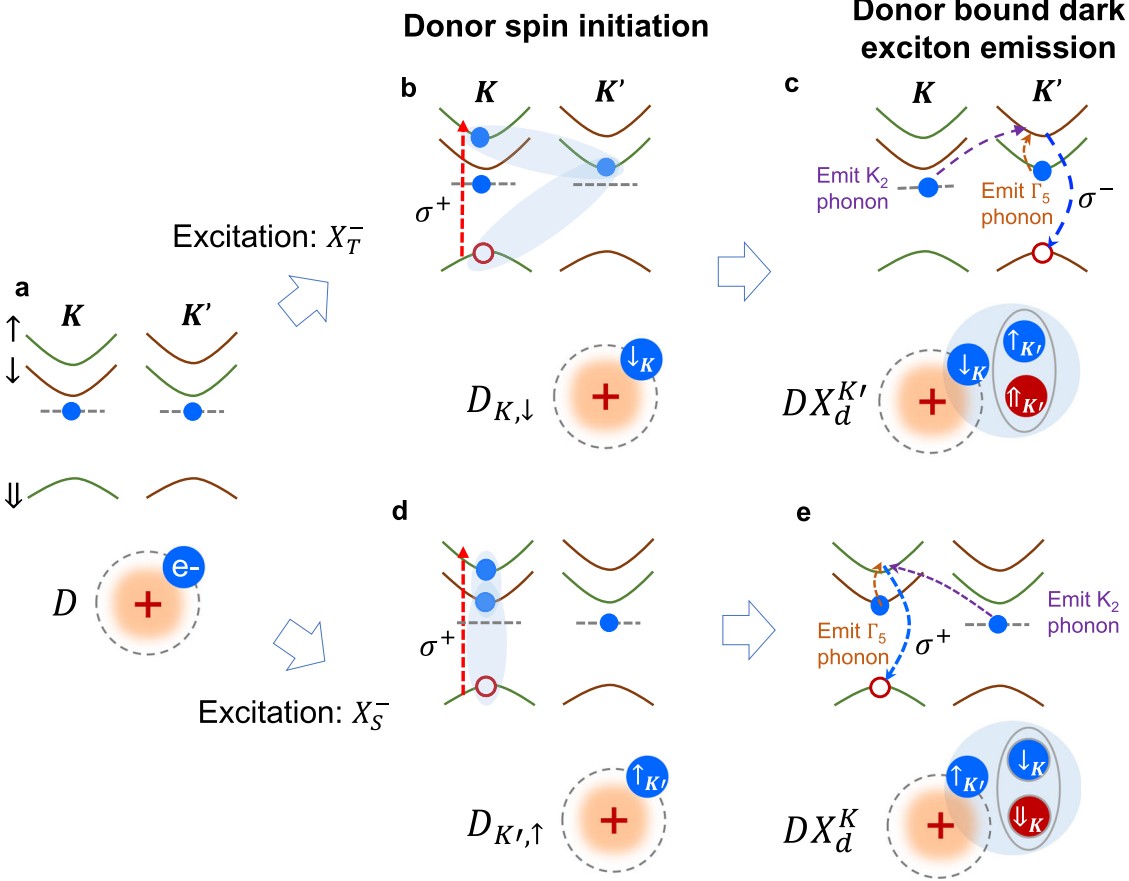

**Fig. 5 Schematic of donor bound dark excitons and spin state initialization. a** Top is a schematic of the spin-valley coupled band edges. Green and brown lines denote electron (hole) spin pointing up (down) and down (up), respectively. Dashed lines indicate a shallow donor level, where carrier's spin-valley locked index is preserved in the relatively smooth trapping potential. Bottom cartoon illustrates the donor configuration $D$ where an electron (blue circle) is trapped by a positive charge center. Under σ+ polarized resonant excitation of free intervalley trion $X_T^-$, **b** the formation of $X_T^-$ (top cartoon) depletes the donor electron spins in $K'$ valley. This results in optical orientation of spin and valley polarized electron coupled to the shallow donor (bottom cartoon, $D_{K,\downarrow}$). $D_{K,\downarrow}$ can then capture a neutral dark exciton in the opposite valley, **c**, forming a donor bound dark exciton $DX_d^{K'}$. $DX_d^{K'}$ can emit light via defect assisted direct electron-hole recombination (peak S1), or via coupling to the bright trion assisted by phonons, which are by either emitting valley conserved but electron spin flipped $\Gamma_5$ phonon (peak S3), or emitting electron spin conserved but valley flipped $K_2$ phonon (peak S2). The opposite hole valley configuration of $X_T^-$ and $DX_d^{K'}$ dictates that the emitted light of $DX_d^{K'}$ is σ- polarized and opposite to that of excitation. **d**, **e** illustrates the scenario of resonant excitation of free intravalley $X_S^-$, which depletes spin down electrons of donor in the same valley. The donor bound dark exciton formed then has the same hole valley configuration, and emit light of the same circular polarization as the excitation. See text for details. Note that the schematics used here are based on single particle picture, for the convenience of explaining spin, valley, and charge degrees of freedom of the quasiparticles, and the stokes process with different phonons.

**Sample fabrication**. The van der Waals heterostructure samples used in this study were fabricated by polycarbonate-based viscoelastic dry-transfer techniques. The polymer residues were cleaned by baths in chloroform and isopropyl alcohol. The individual layers were obtained from the mechanical exfoliation of bulk crystals onto 285 nm of thermally-grown $SiO_2$ on p+ doped Si wafers. The thickness of hBN layers was determined by atomic force microscopy. Monolayers of $WSe_2$ were identified by their optical contrast, which was later confirmed by their low energy PL spectrum.

**Electrostatic doping**. Standard electron beam lithography was used to produce PMMA masks for subsequent electron beam evaporation of V/Au (nominally, 5/50 nm) electrodes to the graphite backgates, as well as to a small portion of the monolayer $WSe_2$ protruding from underneath the top gate dielectric. For fully-encapsulated monolayer $WSe_2$ samples, electrodes were deposited to a second piece of thin graphite that was both in contact with the monolayer $WSe_2$ and also protruding from underneath the top gate dielectric. The applied backgate voltage was controlled by analog output from a National Instruments USB I/O DAQ board using mxdaq drivers in Matlab environment. The gate leakage current was actively monitored using a combination of transimpedance amplifier and analog input on the DAQ board.

**Photoluminescence spectroscopy**. Photoluminescence measurements were performed in a home-built confocal microscope, in reflection geometry, normal to the plane defined by the monolayer $WSe_2$. The samples were either (1) mounted on the cold head of a closed-cycle He cryostat at a temperature of 5 K and studied using an IR-enhanced achromatic 50X objective lens (0.65 NA) or (2) mounted inside a He-exchange-gas cooled cryostat (attocube attoDRY 2100) at a temperature of 1.6 K and studied using an IR-enhanced achromatic 100X objective (0.81 NA) with non-magnetic Ti housing. In both cases, the samples were illuminated by power-stabilized HeNe laser light ($\lambda$ = 1.96 eV) focused to a beam waist of ~1 μm. Circularly polarized excitation and detection was achieved by an appropriate combination of fixed linear polarizers and $\lambda/4$-waveplates, with achromatic $\lambda/2$-waveplates mounted in stepper-motor-controlled rotation stages. Linearly polarized excitation and detection was achieved by an appropriate combination of fixed linear polarizers and $\lambda/2$-waveplates mounted in stepper-motor controlled rotation stages. The collected PL was directed into a 0.5-m spectrometer, where it was dispersed by a 600-line/mm grating with 750 nm blaze before being detected by Si charge-coupled-device. The polarization of the light entering the spectrometer was S-polarized for all polarization-resolved measurements.

**Photoluminescence excitation spectroscopy**. The excitation source was a narrowband (<20 kHz linewidth) and frequency tunable Ti:Sapphire continuous-wave laser ($M^2$ SolsTiS). The laser line was filtered from the collection path using a combination of bandpass filter and tunable long- and short-pass filters (Semrock VersaChrome). The optical power was stabilized by servo control of the power in the first order diffracted beam using feedback control of the voltage applied to the acoustic-optic modulator.

**Time resolved photoluminescence spectroscopy.** Time-resolved PL measurements were performed by directing the collected photons onto an IR-enhanced single-photon avalanche photodiode (Excelitas) connected to a time-correlated single photon counting system (PicoHarp 400). Spectral filtering of the signal was achieved by either (1) a combination of bandpass filter and tunable long- and short-pass filters (Semrock VersaChrome), or (2) using the 0.5-m monochromator to disperse the signal, which was then filtered at the exit port by an adjustable slit assembly. The excitation was provided by spectrally filtered (~2 nm bandwidth at FWHM) output from a supercontinuum fiber laser with 100 kHz repetition rate and ~10 ps pulse duration.

**Scanning tunneling microscopy.** STM measurements were performed using a Scienta Omicron STM system at room temperature under an ultra-high vacuum (base pressure <1.0 × 10-10 torr). $WSe_2$ bulk crystals were mounted onto a metallic sample holder with silver epoxy, and then cleaved in situ in the UHV STM chamber to obtain a clean surface. The tungsten tip was cleaned and calibrated against an Au(111) surface before all the measurements. For each STM image, the defect density was calculated through the number of defects divided by the scanning size. To avoid the localized counting, each value was obtained from the average of 30 ($50 \times 50$ nm$^2$) STM images in different area.

## Data availability
The data that support the findings of this study are available from the corresponding authors upon reasonable request.

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

## Acknowledgements

This work was mainly supported by Army Research Office (ARO) Multidisciplinary University Research Initiative (MURI) program (grant no. W911NF-18-1-0431) and NSF EFRI (Grant No. 1741656). Time-resolved measurements are supported by the Department of Energy, Basic Energy Sciences, Materials Sciences and Engineering

Division (DE-SC0018171). H.M. acknowledges support from a Samsung Scholarship. DE acknowledges partial support from the NSF Center for Integrated Quantum Materials (CIQM). W.Y. and H.Y. were supported by the Research Grants Council of Hong Kong (C7036-17W), and the University of Hong Kong Seed Funding for Strategic Inter-disciplinary Research. D.G.M. and J.Y. were supported by the US Department of Energy, Office of Science, Basic Energy Sciences, Materials Sciences and Engineering Division. WSe$_2$ synthesis and STM characterization were supported by the NSF MRSEC program through Columbia in the Center for Precision Assembly of Superstratic and Superatomic Solids (DMR-1420634). K.W. and T.T. acknowledge the support from the Elemental Strategy Initiative conducted by the MEXT, Japan, Grant Number JPMXP0112101001, JSPS KAKENHI Grant Numbers JP20H00354 and the CREST(JPMJCR15F3), JST. H.D. is supported by the Department of Energy, Basic Energy Sciences, under Award No. DE-SC0014349. X.X. acknowledges the support from the State of Washington funded Clean Energy Institute and from the Boeing Distinguished Professorship in Physics.

## Author contributions

X.X., W.Y., J.H., D.E. and A.P. supervised the projects. P.R and M.H. fabricated the devices, performed optical spectroscopy measurements, and firstly identified the exci-tonic satellites with bulk crystal provided J.Y. and D.G.M.; B.K., S.L. and D.A.R. performed defect control WSe$_2$ crystal synthesis, device fabrication, STM and optical spectroscopy measurements and analysis. C.R.V. performed and analyzed atomically resolution STM/STS measurements. H.M. and L.M. performed super resolution mea-surements. K.W. and T.T. provided high quality hBN crystal. X.X., W.Y., P.R., H.Y., J.H. and H.D. interpreted the results with input from all authors. X.X., P.R., M.H. and W.Y. wrote the paper with inputs from all authors. All authors discussed the results.

## Competing interests

The authors declare no competing interests.
