## [Peer Review File · Nature Communications]

REVIEWERS' COMMENTS

Reviewer #3 (Remarks to the Author):

The authors have satisfactorily addressed the comments and criticism raised in my previous report (for Nature Nanotechnology). In addition, I think the manuscript possesses sufficient advance in the field to warrant publication in Nature Communications.

A minor note. I criticized the assumption that defect state spin is locked to conduction band valley as this was only intuitively justified based on the fact that shallow defect states can be considered to arise from "a relatively smooth trapping potential" and thus inherit the conduction band properties. However, in a recently published paper [Y. Wang et al., Nano Lett. 20, 2129 (2020)], this was explicitly shown by DFT calculations (also) for a deep state, and could be used to solidify the authors' claim.

Reviewer #3 (Remarks to the Author):

The authors have satisfactorily addressed the comments and criticism raised in my previous report (for Nature Nanotechnology). In addition, I think the manuscript possesses sufficient advance in the field to warrant publication in Nature Communications.

A minor note. I criticized the assumption that defect state spin is locked to conduction band valley as this was only intuitively justified based on the fact that shallow defect states can be considered to arise from "a relatively smooth trapping potential" and thus inherit the conduction band properties. However, in a recently published paper [Y. Wang et al., Nano Lett. 20, 2129 (2020)], this was explicitly shown by DFT calculations (also) for a deep state, and could be used to solidify the authors' claim.

Response: We thank the reviewer for suggesting the Nano Letters paper, which we have now cited as Ref. 50 in the revised manuscript.